# Passive Wireless Dual-Tag UHF RFID Sensor System for Surface Crack Monitoring

**DOI:** 10.3390/s21030882

**Published:** 2021-01-28

**Authors:** Peng Wang, Lihong Dong, Haidou Wang, Guolu Li, Yuelan Di, Xiangyu Xie, Dong Huang

**Affiliations:** 1School of Materials Science and Engineering, Hebei University of Technology, Tianjin 300401, China; wangpengzgsd@163.com; 2National Key Laboratory for Remanufacturing, Army Academy of Armored Forces, Beijing 100072, China; Lihong.dong@126.com (L.D.); wanghaidou@aliyun.com (H.W.); dylxinjic031@163.com (Y.D.); 3School of Materials Science and Chemical Engineering, Harbin Engineering University, Heilongjiang 150001, China; xiangyuxie163@163.com; 4College of Mechanical and Electrical Engineering, Jiangxi University of Science and Technology, Jiangxi 341000, China; huangdong2020ye@163.com

**Keywords:** dual-tag, coupling effect, sensing performance, ultra-high frequency (UHF), radio frequency identification (RFID), structural health monitoring (SHM)

## Abstract

The generation and propagation of cracks are critical factors that affect the performance and life of large structures. Therefore, in order to minimize maintenance costs and ensure personal safety, it is necessary to monitor key structures. The sensor based on ultra-high frequency radio frequency identification (UHF RFID) antenna has the advantages of passive wireless, low cost, and great potential in the field of metallic structure health monitoring. In this paper, aimed at the key parts of a metallic structure, a dual-tag system is used for crack monitoring. In conjunction with mode analysis, the principles of the sensing tag and the coupling principles of the dual-tag are analyzed. Considering that the dual-tag is placed in different methods, the effect of mutual coupling on the sensing performance of the tag is studied. The results show that the frequency of the sensing tag can be tuned by adding the interference tag, and the dual-tag sensor system has reasonable sensitivity. The results also provide potential guidance for the optimal placement of multiple tags in the near-field region.

## 1. Introduction

Structural health monitoring (SHM) has become a necessary measure to ensure the safe and reliable operation of large structures (railways, bridges, aircrafts, etc.) [1]. In civil machinery and aviation systems, the failure and fatigue of metallic objects may cause economic loss and endanger personal safety [2]. Therefore, the detection of stress and fatigue cracks are important aspects of in-depth evaluation of sensitive metallic structures [3]. We can detect failures by embedding compact and portable sensors in the structure. The commonly used sensors are strain gauges [4], optical fiber sensors [5], etc. These appropriative sensors need external wiring, which leads to complex systems and the increase in weight. At the present stage, non-destructive testing and evaluation (NDT & E) technology, such as magnetic particle testing (MT) [6], ultrasonic testing [7], eddy current testing [8], thermal imaging/infrared testing [9], and so on, are widely used in metallic detection. These methods have high detection sensitivity and reliability. However, they have the disadvantages of a short detection distance, being time-consuming, their costliness, and so on, which limits their scope of application for SHM.

With the development of sensors and the Internet of Things (IOT), defect detection and characterization techniques will be applied to products and infrastructures [10]. Radio frequency identification (RFID) systems have developed rapidly in the past decades. Because of their passive, low-cost and wireless characteristics, they are used for identification and tracking of items [11]. RFID technology can be used to detect state changes of the studied objects and environment, which has attracted people’s attention. When the tag receives a radio frequency signal which is sent by the reader, there is an induced current on its surface and it sends spectrum signals. After being decoded by the reader, these signals are be transmitted to the system for data processing [12]. It can monitor objects for life and bridge the gap between NDT & E and SHM very well.

People use RFID technology on the basis of identification and tracking, and then give it the function of perception. Among them, tags can perceive the change of object attributes, which becomes a new trend of passive sensing [13]. We can attach the tags on the surface of the detected objects or embed them in the detected objects. According to data information before and after detecting, we can analyze the change of strain or cracks [14]. RFID tags have the advantages of a long reading range, passive operation (without onboard battery), a simple configuration, and good applicability [15]. In the future, passive RFID sensors in SHM have great development potential as a green technology.

Researchers have attempted to use RFID to record and monitor the evolution of cracks in civil and mechanical structures. The presence of a crack causes a resonance frequency of the antenna sensors to shift to the lower frequency as compared to the original one without a crack. RFID tags for crack sensing may be categorized into two main families. The first class of methods detect cracks by physical deformation of the tag antenna or changing field distribution in the covered area during the crack propagation. When the crack extends to a certain point, the bottom copper patch of the tag begins to fracture and gradually propagates to the substrate and the top copper patch. The relationship between resonance frequency and crack can be established to detect the crack opening width [16]. Prasanna [17] proposed a sensor design and the development of a 2D grid of tags that can be used to monitor the position, length, and orientation of a crack. With the development of the crack, it could cut off some antennas in the grid, thus the impedance and radiation characteristics of the tag change permanently. However, there are still some cracks that miss detection in the 2D grid. Martínez-Castro [18] developed a crack detection sensing system with a single RFID tag and a 2D array of RFID tags lain by using backscatter power as the parameter of damage identification, which improved the universality of detection. A single tag sensor has a higher sensitivity to the formation of cracks. The 2D array of tags lain had a low precision of crack detection. However, the delaying transfer ratio and the mutual coupling of tags were not considered. The conductive loss is proportional to the operating frequency when an RFID antenna sensor is installed on the metallic surface. With the increase in operating frequency, scaling down the wavelength [19] and the geometry of the antenna can enhance the spatial resolution for monitoring the cracks [20]. Zhang [21] used the proposed ultra-high frequency radio frequency identification (UHF RFID) sensor system combined with the kernel principal component analysis (PCA) method to achieve the quantification of millimeter crack depths. PCA is a multivariate statistical analysis method based on eigenvector. It can capture the nonlinear structure of data by using orthogonal space with a small number of dimensions. The feasibility of in-situ monitoring was proved by experiments. The changing field distribution in tag coverage area leads to the variation of signal parameters. The correlation between cracks and parameters can be established, which can effectively detect the cracks.

The second class of methods use coupled RFID tags. The crack evolution increases the distance between two parts of the radiating element and changes the mutual coupling [22]. Stefano Caizzzone [23] used two coupled tag antennas to achieve sub-mm resolution in crack detection by establishing the correlation between phase and crack width. However, this method was only sensitive to the cracks between both tag antennas making it limited in the in-situ monitoring.

Above all, multiple tags used for crack detection and characterization have been studied in the existing literature, but the influence of the coupling effect between tags was rarely considered. It is necessary to detect a crack in the coverage area of tag and to layout multiple tags in the actual application environment. It is particularly important to study the direction in which the tags are arranged and how the change of distance among tags affect the sensitivity and reliability of crack detection. This paper shows two tags that are placed in the near-field region of UHF RFID. Through modal analysis, the changing rules of mutual impedance are studied, and the influence factors of frequency shift caused by near-field mutual coupling effect are discussed. The sensitivity and reliability of crack detection is investigated when two tags are placed in the near-field. Section 2 presents the sensing tag setup and wireless transmission. In Section 3, the changing rule of mutual impedance in two coupled tags is analyzed by modal analysis. The experimental research and discussion are carried out in Section 4. The last section gives the corresponding conclusion.

## 2. Sensing Tag Setup and Wireless Transmission

### 2.1. Sensing Tag Setup

When there are cracks in the coverage area of sensitive part of the antenna, its performance parameters change accordingly. RFID tags can be easily turned into a crack sensor by this principle.

In general, the current density in the coverage area of a tag is a non-uniform distribution, which will affect the reliability of the detection with the variation of crack position [24]. The structure and dimensions of an antenna are shown in Figure 1. It is a microstrip antenna structure mounted on a metallic surface. The radiation patch and ground patch are etched on the FR4 substrate which has a relative dielectric constant of 4.4 and is connected by a shorting pin. A microstrip feed line is inserted into the radiation patch to decrease the input impedance. The feed embedding depth *L_inset_* is used to tune the real impedance, and the parameter *L_s_* is used to adjust the imaginary part of the antenna [25]. The IC chip of the tag is Alien Higgs-3. When the frequency is 915 MHz, the input impedance is *Z_chip_* = (27 + j201) Ω. The IC chip is attached between the shorting stub and the radiation patch. Among them, the size of the radiation patch is *L* = 85 mm, *L*_1_ = 78 mm, *W* = 28 mm. The size of the microstrip feeder is set as *L_inset_* = 10 mm, *W_inset_* = 12 mm, *L_S_* = 11 mm, *m* = 4 mm. The length of the ground patch is *n* = 7 mm. The bottom of the substrate without patch can be used for crack sensing. In order to evaluate its sensing performance, the crack depth increases gradually.

### 2.2. Wireless Transmission

When the distance between the reader and tag antennas is fixed, the threshold power *P_R_* of the reader can be expressed as follows [26]:(1)$$P_{R}=\frac{P_{\mathrm{sense}}}{G_{\mathrm{reader}}G_{\mathrm{antenna}}{\left(\frac{\lambda }{4\pi r}\right)}^{2}\tau },$$
(2)τ=1-S112=1-Zchip−Zantenna*Zchip+Zantenna2=4Re(Zchip)Re(Zantenna)Zchip+Zantenna2≤1,
where *P_R_* is the threshold power of the reader, *P*_sense_ is the minimum activation power of the chip. *G*_reader_ and *G*_antenna_ are the gain of the reader antenna and the tag antenna. *λ* is the free-space wavelength. *r* is the distance between the reader and the sensing tags. The power transmission coefficient *τ* accounts for the degree of impedance mismatch. The value is related to the impedance of the chip and the antenna. *Z*_chip_ and *Z*_antenna_ are the impedance of the IC chip and the antenna. *Z^*^*_antenna_ is the conjugate value of the antenna impedance. *Re (Z*_chip_*)* and *Re (Z*_antenna_*)* are the real parts of the IC chip and the antenna impedance, respectively.

## 3. Mode Analysis

The above section described the design and wireless transmission of the sensing tag. In this section, CST Studio Suite 2018 software is used to analyze the current distribution at different crack growth stage, which shows that the designed tag can effectively detect cracks. Through the simulation of both closely placed tags, we can visually see the variation of impedance at different distances. The analysis of simulation provides a theoretical basis for the later experiment.

### 3.1. Crack Sensing Mode Analysis

The radiation and impedance characteristics of the antenna will change when there are cracks on the metallic structure. We can relate a change in the physical parameters to the optimal operating frequency band which is the most suitable for the corresponding reader. When there is only a sensing tag, the reactance part affects the resonant frequency of the antenna. According to Thomson formula:(3)$$f=\frac{1}{2\pi \sqrt{L_{ti}C}},$$
where *L_ti_* is the equivalent inductance, *C* is composed of shunt capacitor *C_P_*, and parasitic capacitance *C*_0_, i.e., *C = C_P_ + C*_0_.

Due to the close distance between tag and metallic surface, there is mutual coupling between the RF currents induced on the antenna and the specimen. The length of current path will become larger than that in the healthy state when a crack appears as shown in Figure 2a. The appearance of a surface crack is equivalent to adding an inductor, which changes the impedance *Z*_antenna_ as shown in Figure 2b. The increase in equivalent inductance *L_ti_* leads to the decrease in frequency *f*.

Figure 3 shows the simulated current distribution on the surface of an aluminum sample which is in a healthy state and a crack-existing state. From the area marked by the red circle in the figure, we can find that the magnitude of the current underneath antenna is not uniform and has higher strength in the center area. In order to make the antenna sensors have better sensitivity, the cracks should be placed in the middle area. The offset of frequency becomes smaller when the crack is not in the middle area. Due to the weaker field strength far away from the middle position, it means that crack with the same size appears in different positions of the covered area, which causes different degrees of perturbation. If there are multiple cracks under the coverage area of the sensing tag, the frequency will shift to the lower region than when there is only single crack. It can also be seen that the current density of the tag is disturbed near the edge of the metallic sample. Therefore, we must consider the influence of edge effect in the practical application environment. The miniaturized antenna can effectively reduce the influence of edge effect and improve the reliability of detection [21]. We also find that the perturbation of current distribution in antenna coverage area is more severe with the increase in crack depth. This change is responsible for the increase in the effective length of the current on the metal surface.

The field distribution is disturbed when there is crack in the surface of the aluminum sample. The surface currents change with the growth of crack depth, which indirectly affects the performance parameters of RFID tags. The reactance of the input impedance will be affected by this perturbation [27]. It can be seen in Figure 4 that the imaginary parts of the impedance shift to the low frequency region, and the real parts of the impedance change slightly with a constant increase in crack depth. The crack will make the equivalent inductance increase. According to Formula (3) and the simulated input impedance at different depths, the frequency f will shift to the lower region, which is consistent with the analysis of crack sensing principle.

### 3.2. Dual-Tag Mutual Coupling Mode Analysis

When the tags are densely placed in the recognition range of the reader antenna, inductive coupling can affect the transmission power and working frequency. The impedance of the sensing tag changes due to the coupling effect of interference tags.

In order to simplify the analysis process, this section takes the dual-tag as an example. The expression of mutual impedance between two tags is [28]:(4)$$\left|Z_{21}\right|=\frac{\left(\left|Z_{t1}+Z_{l1}\right|\left|Z_{t2}+Z_{l2}\right|\right)}{\sqrt{32R_{t1}R_{l1}R_{t2}R_{l2}}}\omega M_{21},$$
where *Z*_11_
*= Z_t_*_1_*+Z_l_*_1_ is the self-impedance of sensing tag1, *Z*_22_
*= Z_t_*_2_*+Z_t_*_1_ is the self-impedance of interference tag 2, and *R_t_*_1_, *R_l_*_1_, *R_t_*_2_, *R_l_*_2_ are the resistance of both tags respectively. *M*_21_ is the mutual inductance.

The mutual inductance *M*_21_ between sensing tag 1 and interference tag 2 is:(5)$$M_{21}=k_{21}\sqrt{L_{t1}L_{t2}},$$

The phase of *Z*_21_ is:(6)$$∠\left(Z_{21}\right)=\phi_{21},$$

*k*_21_ is the coupling coefficient, *L_t_*_1_ is the equivalent inductance of the sensing tag, and *L_t_*_2_ is the equivalent inductance of the interference tag.

When the dual-tag has the same type, the coupling coefficient *k*_21_ is:(7)$$k_{21}=\frac{L_{m}}{\sqrt{L_{t1}L_{t2}}}=\frac{f_{1}^{2}-f_{2}^{2}}{f_{1}^{2}+f_{2}^{2}},$$

The mutual inductance *M*_21_ of sensing tag 1 will also change if the distance between double tags is close, which results in the frequency shift. When the microstrip tags are densely placed, the real part of the sensor tag did not change, and the coupling effect has a great influence on the imaginary part. The mutual impedance of the imaginary part could accurately reflect the strength of mutual coupling. In the RFID near-field system, the imaginary part of mutual impedance between double micro-strip tag antennas is:(8)Im(Z21)=Z212−Re(Z21)2≈Z21=Zt1+Zl1Zt2+Zl232Rt1Rl1Rt2Rl2ωM21,
where *Z*_11_
*= Z_t_*_1_*+Z_l_*_1_ is the self-impedance of sensing tag1, *Z*_22_
*= Z_t_*_2_*+Z_t_*_1_ is the self-impedance of interference tag 2, and *R_t_*_1_, *R_l_*_1_, *R_t_*_2_, *R_l_*_2_ are the resistance of both tags respectively. *M*_21_ is the mutual inductance.

A number of tags need to be placed in a practical working environment (railways, bridges, aircrafts, etc.). For example, 8 and 16 tags can be placed for crack monitoring in the areas of high stress concentration in the airplane wing grid. Due to the limitation of structure, interval distance between tags is very small and their mutual coupling will affect the resonant frequency of the sensing tag. Assuming that the coupling effect of any interference tags on the sensing tags can be linearly superimposed, then the frequency is expressed as follows:(9)$$f=\frac{1}{2\pi \sqrt{L_{ti}C_{ti}}}=\frac{1}{2\pi \sqrt{\sum_{i=1}^{n}\left(L_{ti}+\sum_{j=1,j\neq i}^{n}M_{ij}\right)}C_{ti}},$$
where *L_ti_* is equivalent inductance, *C_ti_* is equivalent capacitance, and *M_ij_* is mutual inductance of tags.
(10)$$M_{ij}=k_{ij}\sqrt{L_{ti}L_{tj}}\left(i,j\in N,i\neq j\right),$$

*k_ij_* is the coupling coefficient.

Crack growth will indirectly change the performance of the antenna sensor. The sensing function can be realized by establishing the correlation between performance parameters and crack. In order to evaluate the universality of the tag sensor, both tags are placed side by side to obtain the changed rule of frequency. We must consider proximity effects of the dual-tag [29]. The incident field direction and polarization direction of the reader antenna are fixed in the designed sensing system. According to the coupled-modes theory [30], current distributions of two very close antennas can be denoted as even and odd modes. Due to the very close proximity, the two tag sensors only sustain the even mode radiation [22]. The impedance of the sensor tag is equal to the sum of self-impedance and mutual impedance. Multiple tags need to be arranged in actual working conditions at the same time. The coupling interference between tags should be minimized, which is conducive to crack detection. The dual-tag is taken as an example to find suitable placement. CST software is used to simulate the impedance and field distribution of two tags which are placed in different methods (parallel placement and reverse placement).

In the sensing system, we should consider that the variation of distance affects sensing performance. The electric field distribution and changing impedance at parallel and reverse placement are simulated in this section. The electric field distribution in both methods is displayed in Figure 5. We can intuitively see that there is a weaker mutual coupling effect at the reverse placement. Figure 6 shows the change of self-impedance and mutual impedance of sensing tag 1 when the dual-tag is placed in different methods. When the operating frequency is 940 MHz, it can be seen that the self-impedance *Z_11_* of sensing tag 1 approximately crossed one color stripe with changing distance and basically remained invariable as shown in Figure 6a–c. This indicates that the interference tag has little effect on the self-impedance of the sensing tag. The real part does not change, and the coupling effect has a great influence on the imaginary part when the dual-tags are densely placed. It can be seen from Figure 6d–f that when the dual-tag is placed in parallel placement, the imaginary part of mutual impedance *Z_21_* increases, and the real part changes slightly with the change of distance, which resulted in the increase in mutual impedance. The mutual impedance when the dual-tag is placed in reverse placement is displayed in Figure 6g–i. It can be noticeably observed that the imaginary part crosses multiple color strips and the increasing amplitude of mutual impedance *Z*_21_ is obviously smaller than that of the dual-tag placed in parallel placement. It shows that the mutual impedance has a smaller value when the dual-tag is placed in reverse placement, which is in line with the design of the sensor system.

## 4. Experimental Studies and Results

### 4.1. Test Setup

The test setup of the UHF RFID sensor system is shown in Figure 7a. Tagformance Pro measurement system was used to measure the power threshold and frequency. The antenna of the reader and UHF tag antenna were both linear polarized. The trajectory of the end of the electric field vector was a straight line. Before the test, in order to eliminate the impact of the loss of the test environment and testing path on the test results, it was necessary to calibrate the test system to ensure the reliability of results. The threshold power and frequency of the tag were measured in a wide frequency range by using the function of threshold measurement. The frequency scanning range of the reader was 800–1000 MHz. The frequency step was 1 MHz and the power step was 0.1 dBm. After parameter setting, the measurement system automatically swept the frequency and power. The distance between reader and tag antenna was 30 cm. A computer was used to process measured data from the reader.

The size of an aluminum alloy 6061 was 200 mm × 100 mm × 5 mm. Four cracks were made by electrical discharge machining (EDM), with an increasing depth from 1 to 4 mm in a step of 1 mm with a fixed width of 1 mm and a length of 50 mm as shown in Figure 7b. An aluminum sample without a crack was used as the initial state of the metallic structure to be measured, and the corresponding crack depth was 0 mm. The prototype antenna sensor was directly placed on the surface of aluminum sample and was perpendicular to the prefabricated cracks. The tag sensor is more sensitive to change in crack depth due to a crack being perpendicular to the direction of installation. As the relative direction between tag and crack changes, the sensitivity becomes worse. A crack depth cannot be quantitatively characterized when a crack is parallel to the direction of installation. The interference tag was placed on the left side of the sensing tag and moved to 20 mm in a 5 mm step length starting from 0 mm apart. The threshold power and frequency of the sensing tag were measured by the reader.

### 4.2. Results and Discussion

With the increase in crack depth, the impedance of the tag sensor can be affected, and the threshold power also changes. The measured value of *P_R_* with different crack depths and the changing resonant frequency in the condition of no interference tag are plotted in Figure 8. It can be observed from Figure 8a that the resonant frequency of the sensing tag shifted towards the low frequency region when the crack propagates. This result is consistent with simulated mode analysis. We noticeably observed that the resonant frequency of the sensing sensor was 970 MHz when there was no crack, which is different from the simulated value of 940 MHz. The main reason was that the soldering between the chip and tag antenna had parasitic capacitance and the surface of the tag antenna was sprayed with a layer of black ink, which caused a shift of resonant frequency. Considering the fabrication error of the antenna, the change of dielectric constant, and permeability, the resonant frequency of the antenna sensor may also shift. It can be seen from Figure 8b that the measured results are nonlinear due to the resolution of the reader and the interference of the wireless channel.

The frequency of sensing tag 1 shifts when two tags are placed side by side. In order to obtain the method of arrangement with relatively little coupling effect, the measure results of the parallel and reverse placement of the two tags are presented here. In the testing process, sensing tag 1 was fixed and the distance *d* between the dual-tag is changed by moving the interference tag. The operating frequency of the sensing tag is measured by the RFID reader. The frequency of sensing tag 1 without a crack in parallel and reverse placement are depicted in Figure 9. As Figure 9a,b show, the offset of the resonance frequency increased with the decrease in distance d. The main reason is that the mutual impedance *M_ij_* and the imaginary part of impedance increased with the decrease in distance. It can be seen from Formula (9) that the increase in mutual impedance *M_ij_* led to the larger offset of frequency. We see from Figure 9c that the difference of offset was 10 MHz with the two placing methods when the distance was the same. The offset of the frequency was much larger when placed in parallel placement. This changing rule is consistent with the simulated variation of impedance and the electric field distribution in Section 3. It can be seen from the experimental results that the frequency of the sensing tag can be adjusted by adding the interference tag, which is used to monitor the existing crack in some structures.

Due to the coupling effect of the interference tag, the impedance mismatch occurred the in the sensing tag, which caused a shift of resonant frequency. When there is a crack in the coverage area of the sensing tag, the increase in the equivalent inductance *L_ti_* also makes the frequency shift to the lower region. The sensing tag is affected by the interference tag and the existing crack at the same time in the design of the sensor system. Figure 10 provides the measured *P_R_* versus frequency with different crack depths at different distances and corresponding frequency characteristics when the dual-tag is placed in parallel placement. As Figure 10a–c show, the resonant frequency is a fixed value when there is no crack in the sample and shifts to the lower frequency region with the increase in crack depth, which conforms to the theoretical formula. The sensing tag can detect cracks well, although there is an interference tag and the resonance frequency continuously shift towards the low frequency region. The relationship between frequency and crack was extracted by a linear fitting technique in order to quantify crack depth. The frequency where the minimum threshold happens was extracted and displayed in Figure 10d, when two tags were placed in different distance. It can be seen that as the crack depth increases by 1 mm, it leads to a 6.5 MHz decrease in the resonance frequency when there is a single tag as sensor. A 1 mm increase in crack depth makes the resonance frequency change by 4.5 MHz when the distance between the two tags is 0 mm. The sensitivity reduces slightly with the decrease in the distance between the two tags and the measured results are nonlinear in different distances. From the mutual impedance Equation (4) and simulation results, it can be seen that the mutual impedance increases with the decrease in distance between two tags, which makes the power transmission coefficient *τ* and the degree of impedance matching decrease. The main reason for the decrease in sensitivity is that the change of distance affects the degree of impedance matching of the sensing tag. The smaller the distance, the effect on the degree of impedance matching is greater and there is lower sensitivity.

Figure 11 plots the measured *P_R_* versus frequency with different crack depths at different distances and corresponding frequency characteristics when the dual-tag is placed in reverse placement. We can find that the sensing tag has a smaller frequency offset when the dual-tag is placed in reverse placement. In other words, this arrangement has weaker mutual coupling effect of the dual-tag, which is consistent with the modal analysis. It can also be seen that with the decrease in the distance between two tags, the sensitivity of crack monitoring becomes slightly worse. The sensing tag can be tuned to make it have a wider frequency range with the addition of an interference tag. In practical application of a multi-tag sensor system, the mutual coupling effect between tags should be small and the appearance of missed detection should be avoided as far as possible. The coupling effect is relatively weak when dual-tag is placed in reverse placement. The results provide potential guidance for further research on the optimal placement of multiple sensing tags. It can be seen from Figure 10d and Figure 11d that the measurement results are nonlinear due to the resolution of the reader and the interference of the wireless channel. The shift of frequency between cracks with a depth of 1, 2, and 3 mm are small. Considering the difference in the air gap between the antenna and the metallic sample caused by the surface roughness, the glue used for tag installation and the generated deviation during the installation process will all affect the resonance frequency. In other words, in order to reduce the influence of these interference factors on the test results, we need to calibrate after installation and strictly follow the operating procedures to reliably detect cracks.

## 5. Conclusions

In the near-field region of UHF RFID, mutual coupling of the dual-tag affects the frequency. In this paper, the dual-tag system was used for crack monitoring. In order to verify the robustness of the dual-tag system for crack monitoring, the influence of distance on sensing performance was studied. In conjunction with mode analysis, we found that the equivalent inductance becomes larger with crack growth in the tag coverage area, which results in the resonance frequency shifting to the low frequency region. The changing distance affects the mutual impedance between the two tags and also cause the frequency to shift. The experimental results show that the designed dual-tag system can be used for crack detection. By adding an interference tag, the installed sensor tag can be calibrated to make them work in the applicable frequency range. In the case of the same distance, the two placement methods have a 10 MHz difference in the offset of the frequency at a healthy state. The offset of the frequency is larger when tags are placed in parallel placement. The sensitivity of the sensor system to crack depth monitoring reduces slightly with the decrease in distance, and the measurement results are nonlinear. The results also provide potential guidance for the future research on the optimal placement of multiple tags in the near-field region.

In this paper, a dual-tag sensing system is established, which considers the case of a crack in the coverage area of a single sensor tag. The miniaturization of the antenna sensor increases the sensitivity of crack monitoring and enables it to be installed in smaller metallic structures. In the future work, we will study the influence of multiple-tag coupling on crack sensing and the changing rules of the sensing performance when the coverage area of the adjacent tag has cracks.

## Figures and Tables

**Figure 1 sensors-21-00882-f001:**
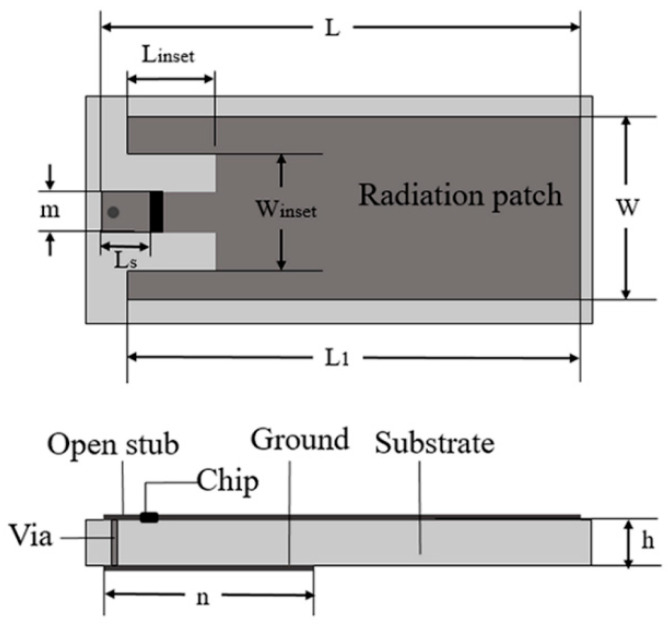
Structure and dimensions of the antenna sensor.

**Figure 2 sensors-21-00882-f002:**
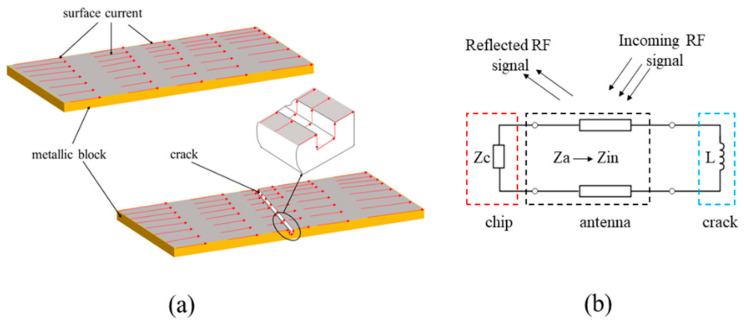
The principles of crack sensing: (**a**) influence of crack on current paths of metallic block and (**b**) the equivalent circuit of antenna.

**Figure 3 sensors-21-00882-f003:**
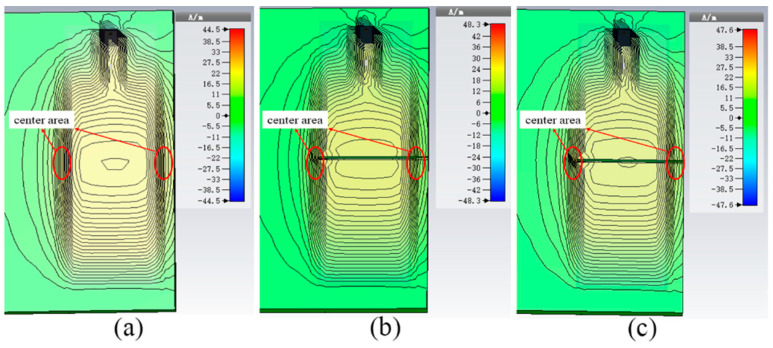
Simulated current distribution on metal surface. (**a**) Healthy state; (**b**) crack depth is 2 mm; (**c**) crack depth is 4 mm.

**Figure 4 sensors-21-00882-f004:**
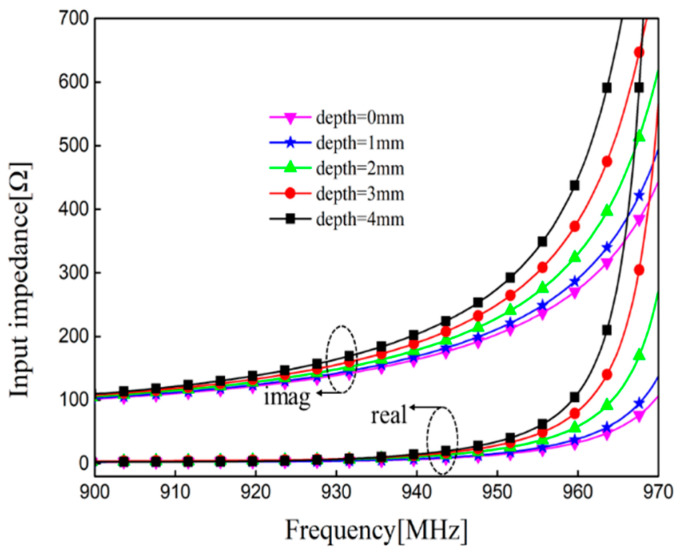
Simulated input impedance in the variations of crack depth.

**Figure 5 sensors-21-00882-f005:**
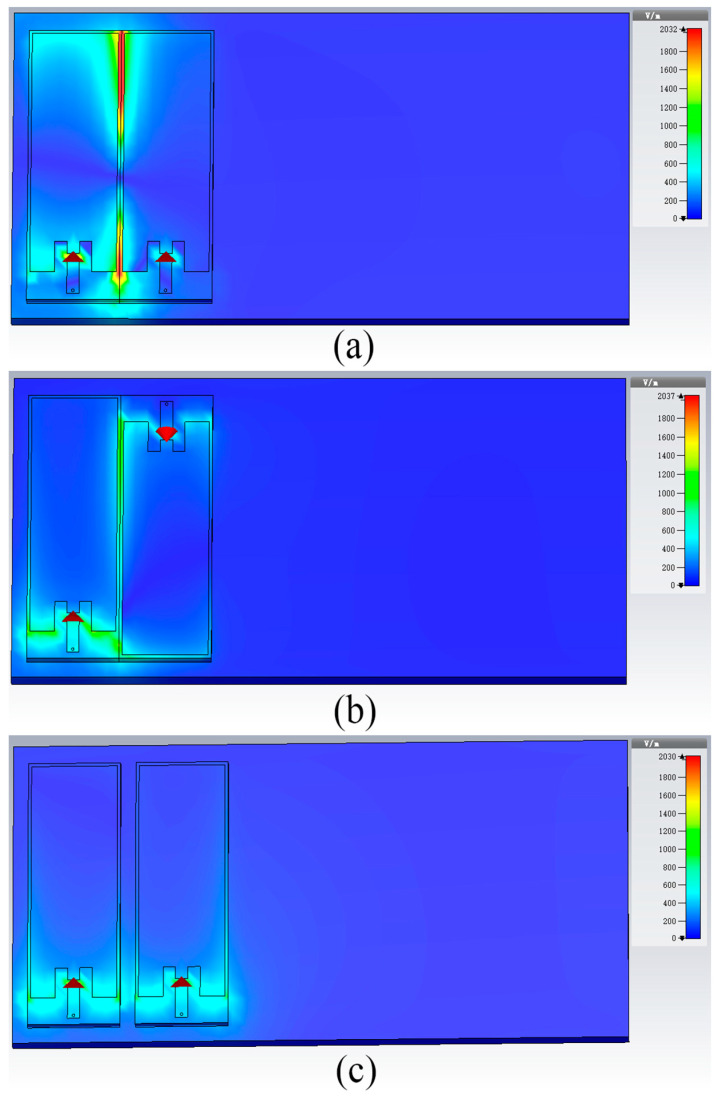
The electric field distribution in both directions: (**a**) the distance is 0 mm in parallel placement; (**b**) the distance is 0 mm in reverse placement; (**c**) the distance is 5 mm in parallel placement; (**d**) the distance is 5 mm in reverse placement.

**Figure 6 sensors-21-00882-f006:**
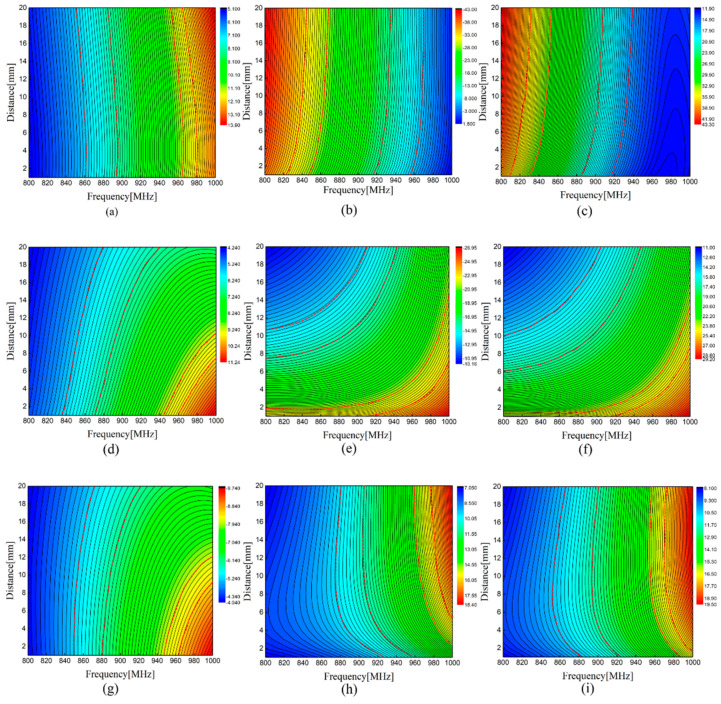
Simulated self-impedance and mutual impedance versus frequency: (**a**) the real part of *Z*_11_; (**b**) the imaginary part of *Z*_11_; (**c**) self-impedance *Z*_11_; (**d**) the real part of *Z*_21_ in parallel placement; (**e**) the imaginary part of *Z*_21_ in parallel placement; (**f**) mutual impedance *Z*_21_ in parallel placement; (**g**) the real part of *Z*_21_ in reverse placement; (**h**) the imaginary part of *Z*_21_ in reverse placement; (**i**) mutual impedance *Z*_21_ in reverse placement.

**Figure 7 sensors-21-00882-f007:**
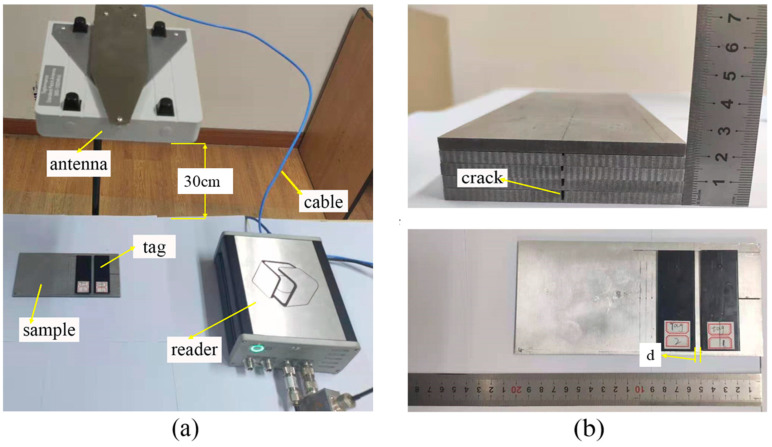
(**a**) Setup of ultra-high frequency (UHF) test system and (**b**) aluminum alloy sample.

**Figure 8 sensors-21-00882-f008:**
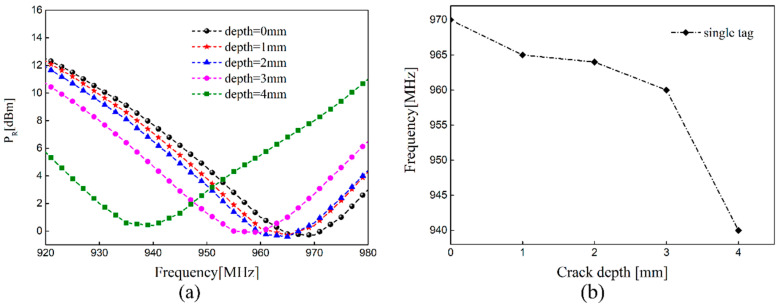
Single tag: (**a**) the measured value of *P_R_* with different crack depths and (**b**) the changing resonant frequency.

**Figure 9 sensors-21-00882-f009:**
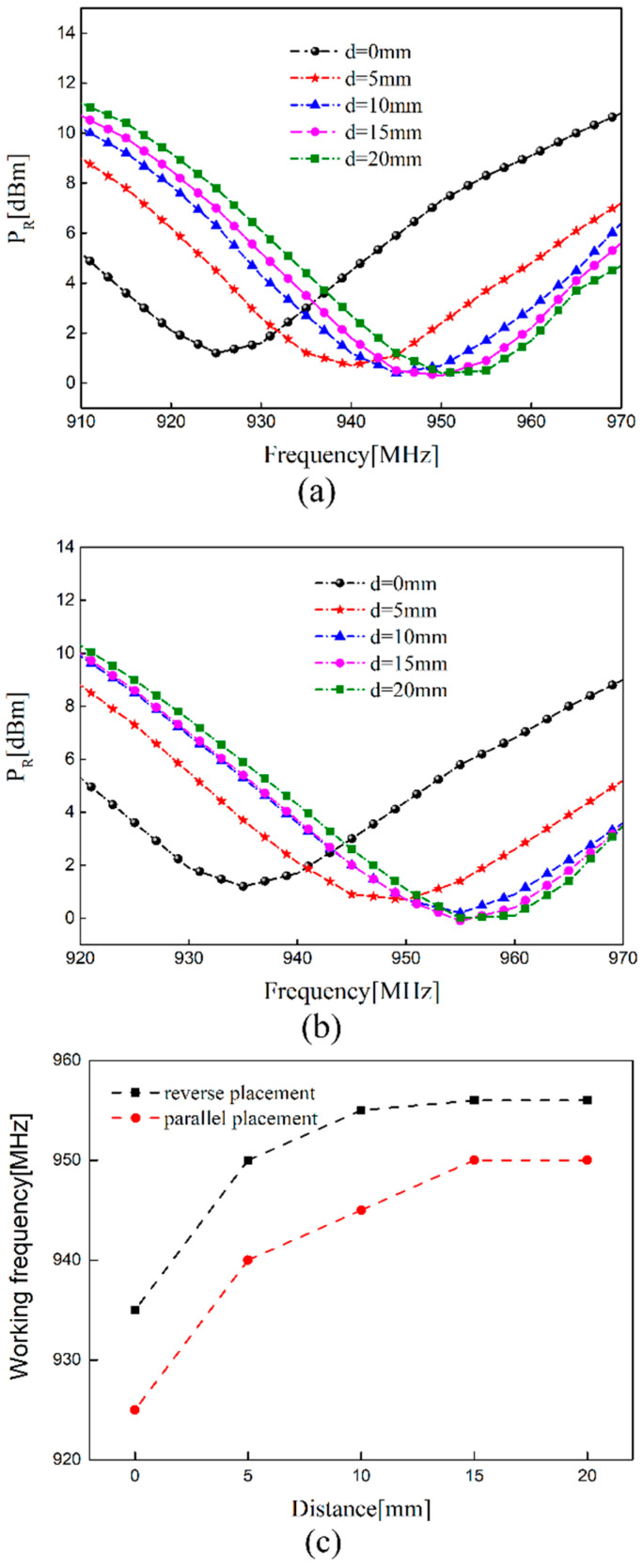
The influence of distance variation: (**a**) *P_R_* versus frequency in parallel placement; (**b**) *P_R_* versus frequency in reverse placement; (**c**) the comparison of working frequency.

**Figure 10 sensors-21-00882-f010:**
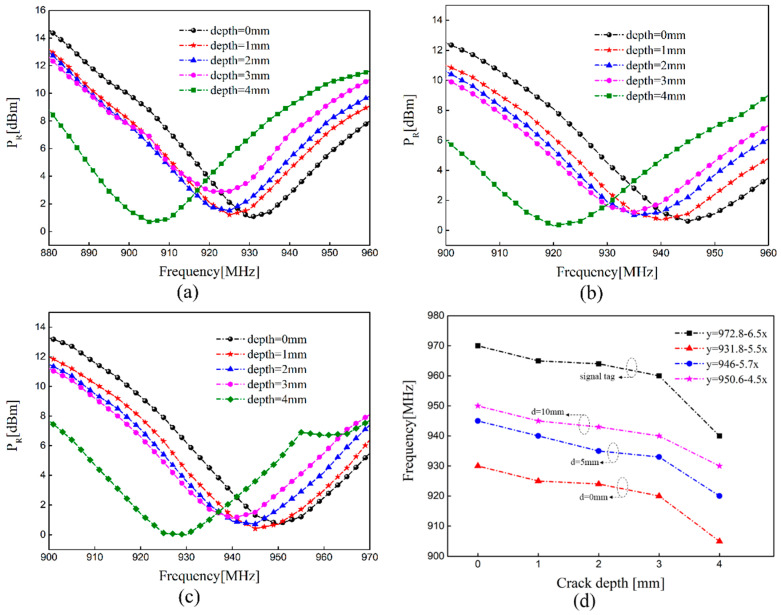
The resonant frequency with different crack depths in parallel placement: (**a**) the distance is 0 mm; (**b**) the distance is 5 mm; (**c**) the distance is 10 mm; (**d**) corresponding features of the resonant frequency at different distances.

**Figure 11 sensors-21-00882-f011:**
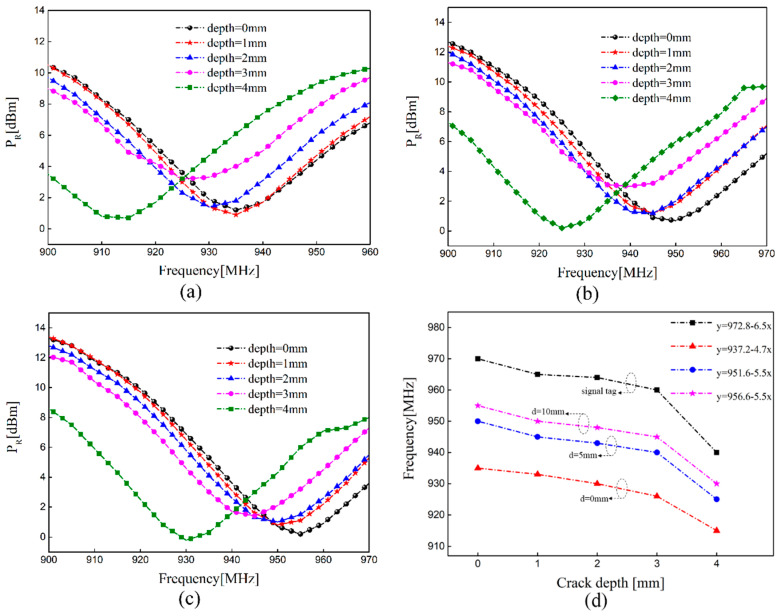
The resonant frequency with different crack depths in reverse placement: (**a**) the distance is 0 mm; (**b**) the distance is 5 mm; (**c**) the distance is 10 mm; (**d**) corresponding features of the resonant frequency at different distances.

## Data Availability

Not applicable.

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
