# Peer review of "Passive Wireless Dual-Tag UHF RFID Sensor System for Surface Crack Monitoring"

_sensors, 2021, doi:10.3390/s21030882_

Round 1

Reviewer 1 Report

This paper discusses the use of RFID sensors for crack detection. Several other groups have produced similar studies and it is not clear how the present work is an improvement over past work. The authors should try to make this more clear.

The authors have produced samples with varying crack depths and then measured the frequency shift on the various samples. Frequently the observed shifts between 1, 2, and 3 mm crack depths were small. See figure 11d, for example. It would be interesting to apply the sensor system to samples with unknown crack depth to see whether the present system could determine the depths. What is the uncertainty in the frequency determinations?

It seems that this method depends upon the relative orientation of the crack and the sensors. How do the authors expect the results to change as the relative orientations are varied? Please present experimental results for this case. 

On line 198 the authors state that a large number of tags would have to be placed in a practical application of this method. Could they please furnish more details of a practical application and state how many tags were used.

Reviewer 2 Report

80: "PCA method"   what is meant by PCA?

120: eqn (2) explain and refer to reference

141: Not clear what authors meant by "concave" and how the surface current "increase" as the crack appears?  Isn't the crack supposed to cut the surface current?

150: "higher strength in center area": please highlight in the figure. 

151: authors tested single crack with different depths. what If there are multiple cracks or a crack but not "placed in the middle area" under the tag? which tag would respond? or all tags will shift?

165: fig 3. what is the substrate material? steel? clarify

how the distribution current increases with crack depth?

169: Fig4 simulated input impedance: which group is real and imaginary? add legend or bundle the real and imaginary groups separately 

200: Authors assumed coupling effect can be "linearly" superimposed

have you verified this by CST or EM simulator? 

219: "the reverse placement" not understood until saw fig 6 two pages later. either explain clearly or consider swap fig 6 before fig 5.

243: fig5: Fix the color scale range for all subfigures, so the value of the variation in (g), (h), (i) can be compared with (d) (e) and (f) 

256: "Tagformance Lite RFID"? 

explain more how the power threshold and frequency are measured? have you swept freq or power or both?  ( in both line 257 and line 296 )

what range and steps of these sweeps?

was it automatic sweep or manual search?

295: distance d... please show on Fig 7b 

299: "change greatly"  do you mean increase or decrease?

322: "relationship between frequency and crack was extracted by the first order curve fitting technique."

curve fitting of what and how?

generally, show how many points used in the plots Fig 8,9,10? can you show the marker points (stars, square, circles ...etc)  on these graphs?

317: "As Figure 10(a)(b)(c) show, the smaller distance between the two tags, the greater offset."

This contradicts what mentioned later in the conclusion (line 364):  

364: "The sensitivity of the sensor system to crack depth monitoring will reduce slightly with the decrease of distance and the measurement results are nonlinear."

317: "As Figure 10(a)(b)(c) show, the smaller distance between the two tags, the greater offset."

Not clear. from Figure 10(a)(b)(c) and clearly in (d), the (offset) variations in all graphs are within 10Mhz for 0 to 3 mm depth. Look very similar

361: "In the case of the same distance, the two placement methods have a 10 MHz difference in the offset of frequency. "

10MHz difference versus what crack depth I guess? then what range of depth?

Could you discuss:

How reliable is the used concept over time? are these figures 8,9,10  reproducible? the resonance of the tag or the reader would shift over time?

Also, In the case of multiple cracks or a crack but away from the tag? which tag from the array will detect the crack? Could you clarify/discuss usage scenario?

Round 2

Reviewer 1 Report

This paper discusses the use of RFID sensors for crack detection. Several other groups have produced similar studies and it was not clear how the present work is an improvement over past work. The authors have tried to make this clear in the introduction. 

The authors have produced samples with varying crack depths and then measured the frequency shift on the various samples. Frequently the observed shifts between 1, 2, and 3 mm crack depths were small. See figure 11d, for example. It would be interesting to apply the sensor system to samples with unknown crack depth to see whether the present system could determine the depths. The authors did not do this, nor did they discuss the uncertainty in the frequency determinations.

It seems that this method depends upon the relative orientation of the crack and the sensors. I requested the authors to explain how the results change as the relative orientations are varied.  This request was ignored. The authors might explain how this would go without doing the experiment, but they did nothing.

On line 198 the authors state that a large number of tags would have to be placed in a practical application of this method. I asked that they please furnish more details of a practical application and state how many tags were used. Nothing happened here either.

Reviewer 2 Report

79: Shift PCA definition after mentioning it.

124: Please add a reference for equation 2 or drive it. It doesn seem to be correct.

Response 3: Not clear the surface current figure with respect to the crack ? Fig 1a in response 3

Response 4: label the red highlighted circle otherwise its confusing the figure. PLease help reader to understand your paper.

Response 5 : add this discussion briefly in the manuscript

Response 15 and 16: do not answer the sensitivity issue. Please clarify this crucial point instead of deleting the sentence. How sensitivity is affected by distance?

Round 3

Reviewer 1 Report

The authors have substantially improved the manuscript and have responded to my previous objections. The manuscript can now be published.